# The Loneliness–Life Satisfaction Relationship: The Parallel and Serial Mediating Role of Hopelessness, Depression and Ego-Resilience among Young Adults in South Africa during COVID-19

**DOI:** 10.3390/ijerph18073613

**Published:** 2021-03-31

**Authors:** Anita Padmanabhanunni, Tyrone Pretorius

**Affiliations:** Department of Psychology, University of the Western Cape, Cape Town 7535, South Africa; tpretorius@uwc.ac.za

**Keywords:** COVID-19 consequences, loneliness, depression, hopelessness, ego-resilience, life satisfaction

## Abstract

Recently, with the onset of the COVID-19 pandemic, several lockdown and stay-at-home regulations have been implemented worldwide. In this regard, loneliness has been identified as the signature mental health consequence of this pandemic. The aim of this study is to explore the associations among loneliness, hopelessness, depression, ego-resilience and life satisfaction in a random sample of young adults (*N* = 337) at a university in the Western Cape of South Africa. Parallel and serial mediation analysis supported the hypothesis that loneliness is associated with hopelessness, which in turn is associated with depression, and that ego-resilience mediates the association between all the negative indices of psychological well-being and life satisfaction. These findings suggest that mental health interventions that boost ego-resilience and target loneliness may help in dealing with the mental health consequences of COVID-19.

## 1. Introduction

Recently, in January 2020, with the outbreak of the COVID-19 pandemic in the city of Wuhan, China, and its rapid global spread, the World Health Organization declared a public health emergency of international concern. With the spread of COVID-19, several governments worldwide have implemented different measures to try to limit the spread of the virus. In South Africa, following the identification of the first case of COVID-19 in March 2020, the government implemented one of the strictest national lockdowns that entailed stay-at-home directives, restrictions on movement, mandatory wearing of face masks, travel restrictions and closure of all non-essential services, including schools, universities, restaurants and shops. In general, both the pandemic itself and the ensuing mitigation policies had a drastic and unprecedented impact on multiple facets of life and precipitated elevated levels of psychological distress among the general public [1,2], including heightened levels of loneliness, anxiety, depression and hopelessness. 

Loneliness is regarded as the signature mental health consequence of COVID-19 [3].

It is an aversive emotional experience arising from a discrepancy between one’s desired and actual social relationships [4] and is characterized by a sense of isolation and a feeling of disconnection from others [5]. Pandemic-related disruptions to social engagement through home confinement, lockdowns and social-distancing protocols have been identified as key factors underlying increased rates of loneliness among the general public. Young adults are particularly vulnerable to loneliness because of their strong reliance on peer relationships, especially when access to such networks may be compromised by the COVID-19 mitigation policies (e.g., home confinement) [6]. This is particularly the case in developing contexts in which digital technologies that can be used to maintain social contact may not be readily available or affordable to a significant portion of the population [2]. 

Loneliness has been positively associated with hopelessness and depression. For example, Snyder [7] conceptualized hope as a positive cognitive schema that influences individuals’ appraisals of themselves, their circumstances and their future. Hopelessness is defined as a system of negative cognitive expectations about oneself, significant others and the future [8]. Prior research suggests that hope is related to improved psychological adjustment in response to health-related crises and lower levels of psychological distress [9]. In contrast, hopelessness enhances the likelihood that life stressors are appraised negatively, diminishes the individuals’ belief in their capacity to cope with adversity and is associated with adverse health outcomes and depression. In the context of the pandemic, it is plausible that the unpredictable course of the disease, high mortality rates and absence of an effective treatment can lead to an increased sense of hopelessness [10].

Loneliness is also a unique predictor of depression [11,12]. According to the socio-cognitive theory of loneliness [13], the experience of being lonely typically motivates people to re-connect with others. However, if social connection is appraised negatively, it can lead to social withdrawal, which can enhance the feelings of isolation and loneliness. In the context of COVID-19, limited social engagement and social withdrawal may be due to pandemic-related restrictions (home confinement, travel restrictions, etc.) and appraisals of social contact as potentially dangerous owing to the threat of contagion. Fear of COVID-19 can also lead to the social exclusion of confirmed survivors and their families [14]. Prolonged social isolation and the absence of pleasurable social engagement and meaningful connections can negatively affect the mood and increase the risk of depression. A low mood, in turn, can influence people’s motivation to engage with others and aggravate social withdrawal and loneliness [15]. 

In addition to its association with hopelessness and depression, loneliness is considered one of the strongest negative predictors of life satisfaction [6,14]. Life satisfaction, a component of subjective well-being, is a cognitive process in which an individual appraises the extent to which they are satisfied with their lives on the basis of self-determined criteria [16]. Social determination theory proposes that life satisfaction is intricately linked to the need for relatedness, that is, having close and meaningful relationships with others. If these relational needs are not sufficiently met, this may lead to loneliness and depression and adversely impact appraisals of life satisfaction [16]. Given the impact of the pandemic and its prevention measures on social engagement, it is possible that relational needs may not be sufficiently met, which may lead to increased levels of loneliness and reduced life satisfaction. 

In sum, loneliness represents the signature mental health consequence of COVID-19 and has been linked to increased levels of depression and hopelessness and reduced life satisfaction. However, not all people experience adverse mental health outcomes during times of crisis, and certain personality characteristics can render an individual more capable of adapting to adversity. Ego-resilience represents one such characteristic, which refers to the ability to flexibly and resourcefully adapt to external and internal stressors [17]. Ego-resilience is a personality trait characterized by a positive temperament and openness to experience. Previous studies [18] have confirmed that individuals with higher levels of ego-resilience can maintain their overall mental health, even during times of crisis. This occurs partly through their experiencing positive emotions more frequently compared to less resilient individuals [19]. 

According to the broaden-and-build theory of positive emotions [20], frequent experiences of positive emotions broaden an individual’s thought–action repertoire and activate personal resources. Furthermore, positive emotions become amplified when they are co-experienced with other people. These shared positive emotional experiences or positivity resonances are a powerful contributor to mental health, beyond the benefits associated with positive affect in general [21]. It is, therefore, plausible that individuals with greater levels of ego-resilience experience more positivity resonances and are able to effectively adapt to the pandemic [19]. 

The aim of the present study is to investigate the parallel and serial mediating role of hopelessness, depression and ego-resilience in the loneliness–life satisfaction relationship. In terms of parallel mediation, the following hypotheses were made:(1)Loneliness is directly and indirectly associated with life satisfaction. In terms of indirect effects, higher levels of loneliness are associated with higher levels of hopelessness, which in turn are associated with lower levels of life satisfaction.(2)With respect to the indirect effects via depression, higher levels of loneliness are associated with higher levels of depression, which in turn are associated with lower levels of life satisfaction.(3)Ego-resilience mediates the relationship between loneliness and life satisfaction.(4)Ego-resilience mediates the relationship between hopelessness and life satisfaction.(5)Ego-resilience mediates the relationship between depression and life satisfaction.

In terms of serial mediation, it has been hypothesized that variables affect each other sequentially. For example, high levels of loneliness lead to high levels of hopelessness, which in turn lead to high levels of depression. Moreover, ego-resilience mediates the relationship between such high levels of depression and life satisfaction. In terms of serial mediation, the following hypotheses were, therefore, made:(6)Loneliness is associated with high levels of hopelessness, and the relationship between hopelessness and life satisfaction is mediated by ego-resilience.(7)Loneliness is associated with high levels of depression, and the relationship between depression and life satisfaction is mediated by ego-resilience.(8)Loneliness is associated with high levels of hopelessness, which in turn are associated with high levels of depression, and the relationship between depression and life satisfaction is mediated by ego-resilience.

## 2. Materials and Methods

### 2.1. Sample

An electronic web-based survey was distributed to undergraduate students during the period of national lockdown from March to June 2020. A sample (*N* = 337) was randomly selected (95% confidence level, 6% confidence interval) from a population of university students. The majority of participants were female (77.2%) and had a mean age of 21.95 years (standard deviation (SD) = 4.7). With reference to COVID-19 status, 82.5% indicated that they had not contracted the virus. A smaller proportion of students either suspected that they had COVID-19 (3.9%) but had not been tested for the disease, or suspected that they had the virus and confirmed this through testing (1.2%). The survey was anonymous, and confidentiality of information was ensured. All the participants provided informed consent. No monetary or other rewards were provided for participation in this survey. Ethical approval for the study was obtained from the Humanities and Social Sciences Research Committee of the University of the Western Cape (ethics code: HS20/5/1). 

### 2.2. Survey Questionnaire 

The first section of the survey investigated the demographic characteristics of the participants and included items pertaining to age, gender and area of residence. The second part of the survey consisted of the following four scales: (a)University of California Loneliness Scale [22], which is a 20-item measure of an individual’s general loneliness and degree of satisfaction with their social network. Responses are measured on a 4-point Likert scale, ranging from 1 (*I never feel this way*) to 4 (*I often feel this way*). Examples of items include *I lack companionship* and *I am no longer close to anyone*. In general, this scale has demonstrated good internal consistency reliability with Cronbach’s alphas ranging from 0.94 to 0.96 [23]. Pretorius [24] reported a Cronbach’s alpha of 0.77 for this scale in South Africa.(b)Satisfaction with Life Scale [25], which is a five-item scale that measures global cognitive judgments of an individual’s life satisfaction. Respondents indicate to what extent they agree or disagree with each of the five items on a 7-point scale, ranging from 1 (*strongly disagree*) to 7 (*strongly agree*). Examples of items include: *In most ways, my life is close to my ideal* and *The conditions of my life are excellent*. Sound internal consistency reliability coefficients for the scale have been reported (α = 0.91) [26].(c)Beck Hopelessness Scale [8], which is a 20-item true/false inventory that assesses the degree to which an individual’s cognitive schemata are associated with pessimistic expectations (e.g., ‘*I don’t expect to get what I really want*’ and ‘*My future seems dark to me*’). Notably, an internal consistency of 0.93 has been reported, along with a concurrent validity of 0.74 with clinical ratings of hopelessness, and 0.60 with other scales of hopelessness [8].(d)Ego-Resilience Scale [17], which is a 14-item measure on a 4-point Likert scale ranging from 1 (*Does not apply*) to 4 (*Applies very strongly*). This scale was created to measure one’s ability to adjust their level of control up or down depending on the circumstances. In general, this scale has shown an acceptable level of internal consistency and positive association with several measures of well-being [27].(e)Center for Epidemiological Studies Depression Scale [28], which is a 20-item depression screening tool that includes seven items on depressed affect, seven items on somatic symptoms, two items on interpersonal problems and 4 items on positive affect. The options for each item range from 0 (*Rarely or none of the time*) to 3 (*All of the time*). The internal consistency coefficients for this scale have been found to range from 0.70 to 0.90 [29].

### 2.3. Data Analysis

IBM SPSS Statistics for Windows (version 26; IBM Corp., Armonk, NY, USA) was used to determine the descriptive statistics and inter-correlations between the study variables as well as reliabilities. Both Cronbach’s alpha and McDonald’s omega were reported for reliabilities as a result of concerns regarding the coefficient alpha under-estimating true reliability in multi-item measurement scales [30,31]. The OMEGA macro, written by the authors of Reference [31] for SPSS, was used for this purpose. 

Structural equation modelling with IBM SPSS Amos (version 26; IBM Corp., Armonk, NY, USA) was used to determine the direct and indirect effects of predictor variables as well as bootstrapping of confidence levels and *p*-values. In contemporary analysis, indirect effects are regarded as a measure of mediation, and their value indicates the amount of mediation. In addition, confidence intervals are used to determine whether the indirect effects are different from zero. If zero does not fall within the confidence interval, the indirect effects are said to be significant [32].

## 3. Results

Table 1 shows the descriptive statistics, inter-correlations and reliabilities for the study variables.

It should be noted that the mean scores obtained for the indices of psychological distress, namely, loneliness (mean (M) = 49.1, SD = 11.6), hopelessness (M = 4.7, SD = 4.4) and depression (M = 27.5, SD = 13.4), are higher than those reported in the literature. For example, in the case of loneliness, mean scores ranging between 36 and 44.9 were reported prior to the pandemic in contexts such as Bangladesh [33], Turkey [34] and the Arab world [35]. These scores are also higher than those reported in a study performed during the COVID-19 pandemic [36]. With respect to hopelessness, the mean score obtained in the current study is also higher than the mean scores reported in other studies [37,38]. Similarly, the mean hopelessness score is also higher than what has been reported in previous research [39,40]. The mean life satisfaction score, however, is lower than what has been previously reported in different contexts, such as Iran [41], India [42], Indonesia and Sweden [43], and lower than those reported during the COVID-19 pandemic [44]. 

All the scales demonstrated satisfactory reliability in terms of both alpha and omega coefficients, which were in many instances identical (both α and ω ranging from 0.82 to 0.93). Moreover, the indices of psychological stress were found to be positively related to each other (*r* = 0.55 to 0.58, *p* < 0.001) and negatively related to life satisfaction (*r* = −0.51 to −0.58, *p* < 0.001) and ego-resilience (*r* = −0.42 to −0.48, *p* < 0.001). This indicates that higher loneliness, hopelessness and depression scores are associated with lower life satisfaction and ego-resilience scores. Ego-resilience has also been found to be positively related to life satisfaction (*r* = 0.44, *p* < 0.001), indicating that higher levels of ego-resilience are associated with higher levels of life satisfaction. 

Figure 1 shows the path analytical model tested and the associated standardized regression weights.

In Figure 1, loneliness is regarded as a predictor and life satisfaction is regarded as an outcome variable. In addition, it is conceptualized that hopelessness, depression and ego-resilience act as both parallel and serial mediators. The results of the path analysis are reported in Table 2 and Table 3. Table 2 outlines the direct effects of the predictor and mediator. 

Table 2 indicates that each of the direct effects were significant because zero falls outside the confidence intervals and *p* < 0.01 at all instances. This largely corroborates the correlational analysis. Loneliness was found to have significant direct positive effects on hopelessness (β = 0.55, *p* = 0.001) and depression (β = 0.39, *p* = 0.001), as well as significant direct negative effects on ego-resilience (β = −0.19, *p* = 0.004) and life satisfaction (β = −0.26, *p* = 0.001). This indicates that higher levels of ego-resilience and life satisfaction are associated with lower levels of loneliness, whereas higher levels of loneliness are associated with higher levels of depression and hopelessness. Similarly, hopelessness was found to have significant direct positive effects on depression (β = 0.35, *p* = 0.002) and significant direct negative effects on ego-resilience (β = −0.28, *p* = 0.001) and life satisfaction (β = −0.29, *p* = 0.001). This means that high scores of hopelessness are associated with high scores of depression, whereas high scores of ego-resilience and life satisfaction are associated with low scores of hopelessness. Depression was found to have significant direct negative effects on ego-resilience (β = −0.15, *p* = 0.003) and life satisfaction (β = −0.14, *p* = 0.010). This indicates that higher levels of ego-resilience and life satisfaction are associated with lower levels of depression. Finally, ego-resilience was found to have significant direct positive effects on life satisfaction (β = 0.13, *p* = 0.003), indicating that high resilience scores are associated with high life satisfaction scores. The indirect effects are reported in Table 3. Given the size of the standardized beta coefficients, it is clear that loneliness was most strongly associated with hopelessness and least strongly associated with life satisfaction. Similarly, hopelessness was most strongly associated with depression and least strongly associated with life satisfaction.

Note that the row numbers in Table 3 correspond to the numbering of the hypotheses in the Introduction Section. Table 3 shows that all the indirect effects are significant as zero falls outside the confidence interval and *p* < 0.05. Thus, the data support our hypotheses. The overall serial mediation model (indirect effect eight) indicates that high levels of loneliness are associated with high levels of hopelessness, which in turn lead to high levels of depression. The role that these indices of psychological distress play in lower levels of life satisfaction is, however, mediated by ego-resilience. 

## 4. Discussion

The aim of the present study was to investigate the parallel and serial mediating role of hopelessness, depression and ego-resilience in the relationship between loneliness and life satisfaction. Several important findings need to be highlighted. First, the levels of loneliness, depression, hopelessness and reduced life satisfaction observed in the current sample were statistically significant in comparison to previous samples from the same population (e.g., Reference [45]) and other normative data (e.g., Reference [37]). This suggests that young adults in South Africa are experiencing unprecedented levels of psychological distress during the COVID-19 pandemic. It is possible that the sudden closure of universities, pandemic-related home confinement, economic volatility, fears of infection and disconnection from peers may have escalated psychological distress among this population [46,47]. 

Second, in line with previous studies [48,49], loneliness strongly predicted depression, hopelessness and reduced life satisfaction. According to the socio-cognitive model, aversive feelings of loneliness typically motivate people to reconnect with others. However, in the context of the current pandemic, social contact with others may not be possible or may be appraised as potentially life-threatening owing to the risk of infection. Such diminished positive social interactions and participation in pleasurable social activities can enhance feelings of loneliness and have a negative impact on mood [15]. Furthermore, reductions in meaningful social activities and engagement in roles that affirm one’s self-concept are associated with a deterioration in well-being, increased feelings of hopelessness and reduced life satisfaction [47]. 

Third, the levels of life satisfaction were found to be comparatively lower than those reported in other contexts during the COVID-19 pandemic. Existing research [50] suggests that, among young adults, social participation through contact with peers is central to enhancing life satisfaction. For the current sample, social participation through class attendance, interaction with peers on campus and organized sports and social activities were impacted by the national lockdown and pandemic-related home confinement. These types of social engagement typically provide young adults with enjoyment and meaning in their everyday life and affirm their social roles [15]. It should be noted that the COVID-19-related restrictions on these activities may account for the low levels of life satisfaction observed among the sample. Furthermore, online and digital technologies that could have been used to promote social engagement are not readily available or accessible in low- and middle-income countries, which may further hamper efforts to stay digitally connected to significant others and may contribute to lower levels of life satisfaction [5]. 

Finally, high levels of loneliness were found to be associated with high levels of hopelessness, which in turn led to high levels of depression and reduced life satisfaction. Ego-resilience was also found to mediate the role of loneliness, depression and hopelessness in lowering the levels of life satisfaction. These findings suggest that loneliness is an antecedent to hopelessness and precipitates depression and reduced life satisfaction. Hence, loneliness should be the central focus in interventions aimed at promoting mental health during the pandemic. The findings also suggest that ego-resilience may be a vital protective mechanism in psychological health. Given that positive emotional experiences with others or positivity resonances are central in both mitigating loneliness and building ego-resilience, increasing these types of social connections may promote mental health. Interventions such as psycho-educational programs delivered through digital technologies or popular media that encourage individuals to create moments of high-quality engagement within existing and accessible social networks may represent one avenue towards achieving this [19]. For individuals residing in rural communities or where there is limited access to digital modes of communication, more rudimentary telecommunication means can be considered for mental health promotion [51].

This study has several limitations. First, a cross-sectional design was used, which limits inferences about causality. Hence, future research should strengthen these findings through a longitudinal design that can test the mediating role of ego-resilience over time. Second, the study relied exclusively on self-reported data from the participants, which can be impacted by recall and social desirability bias. Third, the sample comprised university students only. Therefore, future research should test the hypotheses of this study in the context of other samples. 

## 5. Conclusions

While this study was exploratory in nature and was only associational, it provided some insights into the potential pathways among indices of positive and negative psychological well-being, as well as the role of a protective factor such as ego-resilience. The results supported our hypotheses regarding the direct and indirect effects of loneliness on depression, hopelessness and life satisfaction. In addition, our results confirmed that ego-resilience mediates the association between all the negative indices of psychological well-being and life satisfaction. 

## Figures and Tables

**Figure 1 ijerph-18-03613-f001:**
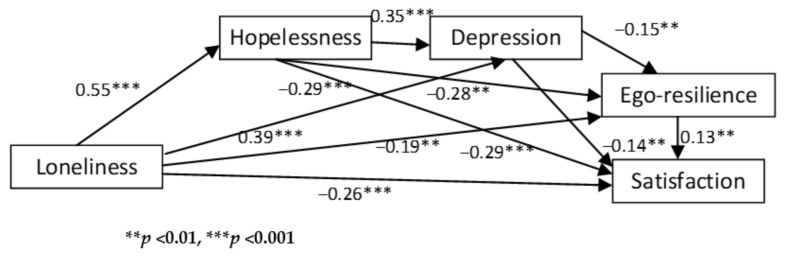
Path analytical model of the inter-relationship between the study variables.

**Table 1 ijerph-18-03613-t001:** Descriptive statistics, inter-correlations and reliabilities for the study variables.

Variable	1	2	3	4	5
1. Loneliness	—				
2. Hopelessness	0.55 **	—			
3. Depression	0.58 **	0.56 **	—		
4. Ego-resilience	−0.44 **	−0.48 **	−0.42 **	—	
5. Satisfaction	−0.56 **	−0.58 **	−0.51 **	0.44 **	—
Mean	49.1	4.7	27.5	41.4	20.2
SD	11.6	4.4	13.4	6.8	7.7
Alpha	0.92	0.88	0.92	0.82	0.89
Omega	0.93	0.88	0.92	0.83	0.89

** *p* < 0.001. SD = standard deviation.

**Table 2 ijerph-18-03613-t002:** Direct effects of the predictor variables and mediators.

Direct Effects	Beta	SE	β	95% CI	*p*
1. Lonely → Hopeless	0.210	0.018	0.551	[0.485, 0.605]	0.001
2. Lonely → Depression	0.444	0.059	0.386	[0.303, 0.474]	0.001
3. Lonely → Ego-resilience	−0.113	0.035	−0.194	[−0.288, −0.094]	0.004
4. Lonely → Satisfaction	−0.174	0.036	−0.264	[−0.350, −0.163]	0.001
5. Hopeless → Depression	1.062	0.144	0.352	[0.268, 0.425]	0.002
6. Hopeless → Ego-resilience	−0.432	0.091	−0.283	[−0.377, −0.186]	0.001
7. Hopeless → Satisfaction	−0.502	0.091	−0.289	[−0.374, −0.199]	0.001
8. Depression → Ego-resilience	−0.077	0.028	−0.152	[−0.242, −0.063]	0.003
9. Depression → Satisfaction	−0.079	0.033	−0.137	[−0.234, −0.042]	0.010
10. Ego-resilience → Satisfaction	0.15	0.052	0.132	[0.056, 0.211]	0.003

Note. Lonely = loneliness; Hopeless = hopelessness; Satisfaction = life satisfaction.

**Table 3 ijerph-18-03613-t003:** Indirect effects of the predictor variables and mediators.

Indirect Effects	B	SE	β	95% CI	*p*
**Parallel Mediation**					
1. Lonely → Hopeless → Satisfaction	−0.105	0.021	−0.159	[−0.139, −0.075]	0.001
2. Lonely → Depression → Satisfaction	−0.035	0.015	−0.053	[−0.062, −0.011]	0.018
3. Lonely → Ego → Satisfaction	−0.017	0.009	−0.026	[−0.034, −0.006]	0.002
4. Hopeless → Ego → Satisfaction	−0.065	0.025	−0.037	[−0.114, −0.030]	0.003
5. Depression → Ego → Satisfaction	−0.012	0.006	−0.020	[−0.025, −0.004]	0.005

**Serial Mediation**					
6. Lonely → Hopeless → Ego → Satisfaction	−0.014	0.005	−0.156	[−0.025, −0.006]	0.003
7. Lonely → Depression → Ego → Satisfaction	−0.005	0.003	−0.059	[−0.011, −0.002]	0.005
8. Lonely → Hopeless → Depression → Ego → Satisfaction	−0.003	0.007	0.194	[−0.006, −0.001]	0.005

Note. Lonely = loneliness; Hopeless = hopelessness; Ego = ego-resilience; Satisfaction = life satisfaction.

## Data Availability

The data presented in this study are openly available in Figshare at doi:10.25379/uwc.13436252.

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
