# Peer review of "The Loneliness–Life Satisfaction Relationship: The Parallel and Serial Mediating Role of Hopelessness, Depression and Ego-Resilience among Young Adults in South Africa during COVID-19"

_ijerph, 2021, doi:10.3390/ijerph18073613_

Round 1

Reviewer 1 Report

First of all, thank you for the opportunity to review this interesting manuscript.

The paper reports an original study whose aim was to investigate the parallel and serial mediating role of hopelessness, depression and ego-resilience in the relationship between loneliness and life satisfaction. The results provide evidence that loneliness is associated with hopelessness, which in turn is associated with depression, and that ego-resilience mediates the association between all the negative indices of psychological well-being and life satisfaction.

The title is appropriate and indicates the main issue of the paper. The abstract provides a complete summery of the content of the manuscript. The key words of the abstract are adequate. The reasons of the study are well-anchored to previous literature and to the specific characteristics of the COVID-19 pandemic. The study employs concepts previously used in the related literature.

The introduction is a good summary of the scientific literature of the concepts used in the study, in the context of the Covid-19 pandemic. It also provides sufficient background information for readers not in the specific field to understand the objectives of the study.

The objectives of the study are clearly and explicitly defined both abstract and in the main text. The methods used are appropriate to the aims of the study. Sufficient information is provided for a capable researcher to reproduce the survey and the statistical analyses, because methods and instruments are presented clearly.

The results are clearly presented in an appropriate format, in particular the tables show essential data that could not be easily summarized in the text. They are also easy to interpret. The number and scientific quality of the references cited are adequate.

In conclusion, I think the article “The Loneliness–Life Satisfaction Relationship: The Parallel and Serial Mediating Role of Hopelessness, Depression and Ego-Resilience Among Young Adults in South Africa During 4 COVID-19” makes an original contribution to the current literature. The study is timely and the manuscript is well written, the research methodology is rigorous, and findings are interesting. Furthermore, I consider the study important because it tested a non-American student sample. While I think that the article should be considered for publication in “International Journal of Environmental Research and Public Health”, I have some recommendations for improving the manuscript. In the following, I will comment on the manuscript page by page – and not by importance of my comments.

Lines 71-72: I would add “negative” before “predictors” to make the sentence clearer.

Lines 140-141: In the Material and Method section it is said that the survey investigated the participants’ demographic characteristics (age, gender and area of residence), but I could not read anything about this in the Results section. Please, add few lines or a table about the demographic characteristics of the sample.

Lines 193-200: at the end of Table 1 **p … , for example **p < .01, should be added.

Lines 241-248: I wonder if the Beta-values are standardized values and which was the strongest direct effect of loneliness. Similarly, which was the the strongest direct effect of hopelessness and depression. May the authors say something about the strongest and weakest effects?

Lines 318-323: May the authors provide more hands-on practical implication of their study?

Line 276: I would say “The aim of the present study was…

Line 334-226: It was good to read that the authors are aware of the limitations of their study, may they also add some future directions of the research in this field?

I wish the authors well in revising the manuscript!

Author Response

Dear Reviewer 1,

Thank you for your valuable feedback. Please see our responses below.

Lines 71-72: I would add “negative” before “predictors” to make the sentence clearer.

Response: "Negative" has now been added, and it now reads:

"In addition to its association with hopelessness and depression, loneliness is considered one of the strongest negative predictors of life satisfaction"

Lines 140-141: In the Material and Method section it is said that the survey investigated the participants’ demographic characteristics (age, gender and area of residence), but I could not read anything about this in the Results section. Please, add few lines or a table about the demographic characteristics of the sample.

Response: we have now included the demographic information under this section see below:

The majority of participants were female (77.2%) and had a mean age of 21.95 years (SD = 4.7). With reference to COVID 19 status, 82.5% indicated that they had not contracted the virus. A smaller proportion of students either suspected that they had COVID 19 (3.9%) but had not tested for the disease; or suspected that they had the virus and confirmed this through testing (1.2%). 

Lines 193-200: at the end of Table 1 **p … , for example **p < .01, should be added.

Response: this has now been indicated

Lines 241-248: I wonder if the Beta-values are standardized values and which was the strongest direct effect of loneliness. Similarly, which was the the strongest direct effect of hopelessness and depression. May the authors say something about the strongest and weakest effects?

Response: The Table contains both standardized (β) and unstandardized  (Beta) coefficients. We have now added lines 265-268 about strongest and weakest effects. It now reads:

Given the size of the standardized beta coefficients it is clear that loneliness was most strongly associated with hopelessness and least strongly associated with life satisfaction. Similarly, hopelessness was most strongly associated with depression and least strongly associated with life satisfaction.

Lines 318-323: May the authors provide more hands-on practical implication of their study?

Response: we have detailed the implications of the study and possible interventions in the Discussion section (from line 324 - 332)

Hence, loneliness should be the central focus in interventions aimed at promoting mental health during the pandemic. The findings also suggest that ego-resilience may be a vital protective mechanism in psychological health. Given that positive emotional experiences with others or positivity resonances are central in both mitigating loneliness and building ego-resilience, increasing these types of social connections may promote mental health. Interventions such as psycho-educational programmes delivered through digital technologies or popular media that encourage individuals to create moments of high-quality engagement within existing and accessible social networks may represent one avenue towards achieving this 

Line 276: I would say “The aim of the present study was…

Response: This has been corrected

Line 334-226: It was good to read that the authors are aware of the limitations of their study, may they also add some future directions of the research in this field?

Response: we have indicated that a longitudinal study on the role of ego-resilience would represent an important future direction  (line 337) and it would be important to investigate the indices of psychological distress among other samples (line 341-342). It reads:

Hence, future research should strengthen these findings through a longitudinal design that can test the mediating role of ego-resilience over time. Second, the study relied exclusively on self-reported data from the participants, which can be impacted by recall and social desirability bias. Third, the sample comprised university students only. Therefore, future research should test the hypothesis of this study in the context of other samples.

Reviewer 2 Report

This article offers data about the loneliness-life satisfaction relationship among young adults in South Africa during COVID19 pandemic.

It is very interesting topic and it is well introduced and the methods selected are also appropriate in this pilot study.

However we detected some concerns that need to be reviewed:

1) the most important is that we do not see any approval by ethics committee, only that informed consent was used and that it was anonymous online survey.

2) The hypotheses are presented more in a dissertation style than a scientific paper. Authors do not refer to these hypotheses when they do the conclusions section which is limited to a couple of lines. Results for each one are described before. A better summary in the conclusions section could be helpful. 

3) The fact that this was not a longitudinal study is a weak point and makes me think about the value of of the research outcome if it can be associated to an specific point in the life of the participants rather than adapting to the pandemic situation.

4) Where is the demographic data for the sample?? no table?

5) Discussion section should be reviewed in depth since it looks like an extension of the results section. Here authors need to go deeper into the possible implications of the findings and in regard of existing literature in the topic which is now available.

6) Tools are well described.

7) There are repeated sentences (eg. discussion with hypotheses section)

8) Last sentence in the results section describes really good the results obtained.

Author Response

Dear Reviewer 2,

Thank you for your valuable input. Please see our responses below

1. We have included information about ethics under the section on Sample. We have stated that:

Ethical approval for the study was obtained from the Humanities and Social Sciences Research Committee of the University of the Western Cape  (line 141).

2. We have provided a summary under the Results section (lines 280-284). In addition, we have added a statement in the Conclusion section indicating that all the hypothesis were confirmed (lines 347-351). In the Results it states:

Thus, all of the hypotheses were confirmed. The overall serial mediation model (indirect effect eight) indicates that high levels of loneliness are associated with high levels of hopelessness, which in turn lead to high levels of depression. The role that these indices of psychological distress play in lower levels of life satisfaction is, however, mediated by ego-resilience.

In the conclusion it states:

The results confirmed our hypotheses regarding the direct and indirect effects of loneliness on depression, hopelessness and life satisfaction. In addition, our results confirmed that ego-resilience mediates the association between all the negative indices of psychological well-being and life satisfaction.

3. In the limitations section of the paper, we had commented on the need for a  longitudinal design to test the mediating role of ego-resilience over time (see lines 337-342). It reads:

Hence, future research should strengthen these findings through a longitudinal design that can test the mediating role of ego-resilience over time. Second, the study relied exclusively on self-reported data from the participants, which can be impacted by recall and social desirability bias. Third, the sample comprised university students only. Therefore, future research should test the hypothesis of this study in the context of other samples.

4. We have included the gender profile and mean age of the participants in the study (line 135), as well as their self-reported COVID-19 status. It now reads:

The majority of participants were female (77.2%) and had a mean age of 21.95 years (SD = 4.7). With reference to COVID 19 status, 82.5% indicated that they had not contracted the virus. A smaller proportion of students either suspected that they had COVID 19 (3.9%) but had not tested for the disease; or suspected that they had the virus and confirmed this through testing (1.2%). 

5. We have discussed the implications of the findings in (lines 324-332). It reads:

Hence, loneliness should be the central focus in interventions aimed at promoting mental health during the pandemic. The findings also suggest that ego-resilience may be a vital protective mechanism in psychological health. Given that positive emotional experiences with others or positivity resonances are central in both mitigating loneliness and building ego-resilience, increasing these types of social connections may promote mental health. Interventions such as psycho-educational programmes delivered through digital technologies or popular media that encourage individuals to create moments of high-quality engagement within existing and accessible social networks may represent one avenue towards achieving this

7. We have linked the discussion to the hypotheses section to allow for a better exposition of the results in relation to the study hypotheses. Hence, this may appear repetitious.

Round 2

Reviewer 2 Report

Authors have done a good job attending the recommendations. However for the ethics committee approval it would be better if they could provide the code or reference of the report.

Author Response

1. The ethics code for the research is HS20/5/1. We have now stated it in the manuscript.